# Epistasis mediates the role of negative frequency-dependent selection in bacterial strain structure

Martin Guillemet[1,2,3]*, Sonja Lehtinen[1,2,3]

1 Department of Environmental System Science, Institute for Integrative Biology, ETH Zürich, Zürich, Switzerland, 2 Swiss Institute of Bioinformatics, Lausanne, Switzerland, 3 Department of Computational Biology, University of Lausanne, Lausanne, Switzerland

* martin.guillemet@agrocampus-ouest.fr

## Abstract

Strain structure is a well-documented phenomenon in many pathogenic and commensal bacterial species, where distinct strains persist over time exhibiting stable associations between genetic or phenotypic traits. This structure is surprising, particularly in highly recombinogenic species like *Streptococcus pneumoniae*, because recombination typically breaks down linkage disequilibrium, the non-random association of alleles at different loci. Recent work suggests that multi-locus negative frequency-dependent selection (NFDS) acts to maintain allelic diversity across bacterial genomes, a pre-requisite for the existence of patterns of linkage disequilibrium. Here, using modeling and genomic analysis, we show that multi-locus NFDS can also shape bacterial strain structure through epistatic effects between these loci. We develop models of two NFDS mechanisms – metabolic niche differentiation and competition-colonisation trade-offs – and show how they can produce epistasis. Notably, both models generate frequency-dependent epistasis. Unlike classical constant sign epistasis, this acts to either reinforce or weaken existing linkage disequilibrium, making observed allele associations contingent on the evolutionary history of the population. We then use a dataset of over 3000 *S. pneumoniae* genomes to test our model predictions, and make observations consistent with frequency-dependent epistatic effects on gene associations. Our results extend and generalise previous theoretical work on the role of antigen-specific acquired immunity (a diversity-maintaining mechanism) on allele associations. Overall, this work contributes to a better understanding of the evolutionary processes shaping the structure of bacterial populations, which is central to predictive modeling of multi-strain pathogens.

provided the original author and source are credited.

**Data availability statement:** The genome data used in this work is publicly available and can be accessed as NCBI BioProjects PRJEB2357, PRJEB2393, PRJEB2395, PRJEB2479, PRJEB2480, PRJEB2417, PRJEB2632. All relevant code has been deposited on the repository: https://doi.org/10.5281/zenodo.17471803.

**Funding:** The study was funded by a Swiss National Science Foundation (SNSF) grant to SL. The grant number is PR00P3_201618. Funder website: https://www.snf.ch/en. The funders had no role in study design, data collection and analysis, decision to publish, or preparation of the manuscript.

**Competing interests:** The authors have declared that no competing interests exist.

## Author summary

Many bacterial species maintain distinct strains with stable genetic associations despite frequent recombination, which should break them down. We propose that mechanisms known to maintain genetic diversity also shape strain structure. Using epidemiological models and genomic analysis of a large *Streptococcus pneumoniae* dataset, we show that diversity-maintaining mechanisms combining across multiple genes can readily generate epistasis. Furthermore, this epistasis that can be frequency-dependent: it either reinforces existing associations, making structure contingent on history, or actively dismantles them. This contrasts with classical constant epistasis. Our findings provide a general mechanism linking the maintenance of diversity to the maintenance of its structure, contributing to the persistence of distinct strains in recombining bacteria and highlighting the role of evolutionary history in shaping pathogen populations.

## Introduction

Many pathogenic and commensal bacterial species exist as distinct strains that are stable through time, a phenomenon referred to as strain structure. These strains can vary significantly in their phenotypic traits, such as immunological characteristics, antibiotic resistance, competitive abilities, or average duration of carriage [1–5]. For instance, certain strains of *Streptococcus pneumoniae* are more likely to cause invasive disease, impacting vaccine design and implementation [3]. Similarly, hyperinvasive clonal complexes in *Neisseria meningitidis* have been observed to persist and spread globally over several decades [1,2]. Strain structure is characterised by strong patterns of linkage disequilibrium (LD) over the whole genome, meaning that pairs of alleles tend to be found together more or less often than expected [6]. This leads to stable associations between phenotypic traits. For example, resistances to different antibiotics tend to co-occur [5], and pneumococcal strains characterised by a longer duration of carriage are more likely to be resistant [4]. Characterising the mechanisms leading to stable associations between phenotypes of interest is important for understanding the ecology of these traits and the dynamics of bacterial strains.

In the absence of recombination, widespread linkage disequilibrium and strain structure are fundamental consequences of clonal reproduction [6]. However, recombination breaks down this structure by shuffling alleles between different genetic backgrounds, thus eroding non-random associations between loci. Over time, in the absence of other factors, recombination is expected to lead to linkage equilibrium. The recombination rate required for strain structure to be eroded in the presence of genetic drift is not entirely clear. This also raises the question whether observed cases of linkage equilibrium arise because of recombination or because of selective forces which maintain a diversity of allele combinations. Nevertheless, especially in highly recombinogenic species such as *S. pneumoniae* [7], we would expect stable

associations between accessory genes or core gene alleles to require active maintenance by selective forces – more specifically epistasis.

Epistasis refers to the interaction between genes at different loci, where the effect of one allele (or gene) on a phenotype is affected by the presence of one or more other alleles: the combined effect is different from the sum of the individual allele effects. Epistasis can stem from molecular interactions, for example with compensatory mutations that alleviate the cost of antibiotic resistance genes [8]. It can also arise from effects on distinct life-history traits because of how these life-history traits combine in the expression of fitness. For instance, antibiotic resistance is expected to be more advantageous to strains that have a long duration of carriage [4,9], and for most vaccine types, emerging resistance is expected to co-occur with virulence alleles [10]. More broadly, however, the extent to which observed allele associations across bacterial genomes arise from epistasis and the mechanisms giving rise to this epistasis are not fully understood.

Here, we explore the role of multi-locus negative frequency-dependent selection (NFDS) in giving rise to epistasis and in shaping bacterial strain structure. The work is motivated by the detection of widespread NFDS acting across bacterial genomes [11–13]. NFDS is a form of balancing selection which promotes allelic diversity through rare alleles having a selective advantage and thus maintains accessory genes at intermediate frequencies. The specific mechanisms maintaining the accessory genome are not fully characterised, and plausibly include a combination of NFDS and other forms of balancing selection (e.g., spatially or temporally fluctuating selection). Here, we specifically focus on NFDS mechanisms–the impact of fluctuating selection on allele associations has been studied previously (e.g., [5]). In epidemiological models, NFDS can arise through antigen-specific acquired immunity [14–16]; within-host niche differentiation (typically conceptualised as metabolic niches) [17] and trade-offs related to direct bacterial competition [18]. We are not aware of other mechanisms giving rise to NFDS in epidemiological models, although additional mechanisms may well exist. Note that here we use the term NFDS consistently with previous literature [11], which, in the case of acquired immunity, may not be strictly accurate [19] because the fitness of variants depends on their historical rather than current frequency. Previous modeling work has suggested NFDS acts to maintain strain structure when combined with asymmetric recombination favouring gene deletion over gene acquisition, through the emergence of outbreeding depression [20]. In this work, we suggest NFDS also shapes strain structure more directly, through epistasis, in particular frequency-dependent epistasis.

Previous models of multi-locus NFDS [11,20] have generally assumed selection acting on multiple loci combines additively. However, because NFDS is a complex form of selection involving feedback from the population composition, it is unclear what this assumption of additivity means biologically and whether it should hold, particularly for functionally related loci. For example, NFDS on antigenic alleles arises because antigen-specific acquired immunity makes transmission to previously colonised hosts more difficult, disadvantaging historically frequent antigens [21]. When considering multiple loci, how this NFDS combines across the loci depends on the relative strength of the immune response to strains carrying one versus two previously encountered antigens. There is no *a priori* reason to assume this should be additive. Indeed, there is considerable theoretical work exploring the effects of different assumptions on the associations between antigenic alleles [22–27], with recent work clarifying how the strength and specificity of cross-protection determine whether antigenic alleles are in LD [27].

Here, we expand and generalise these ideas beyond the role of antigenic loci in generating strain structure, to how NFDS mechanisms more generally impact allele associations. We show that multi-locus NFDS can – depending on how it combines across loci – either abolish or maintain strain structure through emerging frequency-dependent effects on allele combinations. We explore two multi-locus epidemiological models with different mechanisms of NFDS, metabolic niche differentiation and competition-colonisation trade-offs, to demonstrate how these effects arise. We then test specific predictions from these models in a dataset of more than 3000 pneumococcal isolates, showing patterns consistent with the frequency-dependent effects we predict.

## Results

### A simple model of multi-locus negative frequency-dependent selection

We begin by revisiting linkage disequilibrium in the multi-locus NFDS model from Corander et al. [11], in which NFDS combines additively across loci. Our goal here is not to provide new results, but rather to build intuitions and introduce concepts and terminology important to understand the rest of the paper.

This model considers a population of bacteria with two bi-allelic loci, resulting in four possible genotypes $G=\{ab, Ab, aB, AB\}$. The dynamics of each genotype are governed by logistic growth with additive effects from NFDS at each locus. The growth rates (in the absence of recombination) of the genotype with alleles $i$ and $j$ are given by:

$$r_{ij} = r\left(1 - \frac{X_{\text{tot}}}{\kappa}\right) + \rho(p_i^* - p_i) + \rho(p_j^* - p_j)$$

(1)

Here $r$ is the intrinsic growth rate, $\kappa$ is the carrying capacity, $p_x$ is the frequency of allele $x$, $p_x^*$ is the allele frequency favoured by selection and $\rho$ is the strength of NFDS (see Table 1 for the list of parameters). Under this model, NFDS acts independently on each allele, favouring alleles that are below their optimum frequency and disadvantaging those above it. This mechanism ensures that allele frequencies are maintained at an intermediate frequency, but is not expected to affect strain structure in the presence of recombination [20].

To formalise this idea, we quantify LD using $D$:

$$D = p_{AB} - p_A p_B$$

(2)

Thus, $D=0$ indicates random association between alleles, while positive $D$ indicates over-representation of $AB$ and $ab$, and negative $D$ indicates over-representation of $Ab$ and $aB$. Note that we use the metric $D$ for derivations, but prefer to use the metric $D'$ in figures as it is more readily interpretable and always bounded by $-1 \leq D' \leq 1$ independently of allele frequencies, see Methods. To understand the dynamics of LD, it is helpful to also define the individual additive selection coefficient [9] on allele A and B ($w_A$ and $w_B$) and an epistatic term ($w_{AB}$) as:

$$w_A = r_{Ab} - r_{ab}$$
$$w_{AB} = r_{AB} - r_{Ab} - r_{aB} + r_{ab}$$

(3)

Introducing recombination, the dynamics of LD are given by [27]:

$$\frac{dD}{dt} = \sum_{G \in \{A,B\}} D\, w_G(1 - 2p_G) + w_{AB}(p_A p_B + D)((1 - p_A)(1 - p_B) - D) + \Delta_\sigma$$

(4)

where $\Delta_\sigma$ is the net effect of recombination in the population on $D$. In the case of unbiased recombination, this is of sign opposite to current LD [28]. Note how the first term vanishes when the individual selection coefficients at each locus go to zero, i.e., when allele frequencies are at equilibrium. Finally, due to the bounds on the values of D, the term multiplying $w_{AB}$ is $\geq 0$. Thus, when individual allele frequencies are at equilibrium, the sign of $w_{AB}$ drives the build-up of negative or positive LD.

Turning back to our model of multi-locus NFDS, when allele frequencies are optimal, the growth rate is the same for all genotypes (Equation 1), and thus epistasis $w_{AB}=0$. In other words, in the absence of recombination, this model predicts neither increasing nor decreasing effects on LD. Introducing recombination into the model (see Methods, section Multi-locus NFDS), we get $\Delta_\sigma = -\sigma D$ where $\sigma$ is the recombination rate. Recombination drives $D$ towards 0 over time,

**Table 1. List of parameters, their dimensions, and the values or ranges used in figures.** The epidemiological parameters correspond to a basic reproduction number of 3 (in the plausible range for *S. pneumoniae* [44]). However, the specific values are not important: the key results relating the k parameters to the effect on strain structure (Equations 5 and 8) do not depend on the epidemiological parameters. The bounds on the k parameters (0,1) arise from the assumption that co-colonisation is not as efficient as primary colonisation, in line with empirical findings (e.g., [33]).

| Symbol | Description | Dimension | Value(s) in figures |
|---|---|---|---|
| **Simple NFDS Model** | | | |
| $r$ | Intrinsic growth rate | Time$^{-1}$ | |
| $\kappa$ | Carrying capacity | Population density | |
| $p_x^*$ | Allele frequency favoured by selection | Dimensionless | |
| $\rho$ | Strength of NFDS | Time$^{-1}$ | |
| **Mechanistic models** | | | |
| $b$ | Birth rate | Time$^{-1}$ | 4 |
| $d$ | Death rate | Time$^{-1}$ | 1 |
| $\beta_0$ | Baseline transmission rate | (Population density· Time)$^{-1}$ | 2 |
| $\gamma$ | Clearance rate | Time$^{-1}$ | 2 |
| $\sigma$ | Recombination rate | (Population density· Time)$^{-1}$ | $0 - 0.05$ |
| $q$ | Scaling factor for transmission from co-colonized hosts | Dimensionless | $0 - 1$ |
| **Metabolic Niche Model** | | | |
| $k_0, k_1, k_2$ | Co-colonization efficiency parameters | Dimensionless | $0 - 1$ |
| | | | $k_0 < k_1 < k_2$ |
| **Competition-Colonization Trade-off Model** | | | |
| $m$ | Benefit associated with the colonizing allele | Dimensionless | 0.8-2.2 |
| $k_\Delta$ | Co-colonization efficiency parameters | Dimensionless | $0 - 1$ |
| | | | $k_{-2} < k_{-1} < k_0 < k_{+1} < k_{+2}$ |

confirming that in this model, NFDS combining additively across loci does not sustain strain structure under this recombination scheme.

## Epidemiological models of NFDS give rise to frequency-dependent epistasis

It is unclear what the assumption about NFDS combining additively across loci in the phenomenological model means biologically. To understand this better, we now turn to epidemiological models of multi-locus NFDS where the diversity-maintaining mechanism is modeled explicitly. Specifically, we model two mechanisms of NFDS (Fig 1): within-host metabolic niches and a competition-colonisation trade-off. We use these models to explore how assumptions about the underlying biology impact how NFDS combines across loci and how this shapes strain structure.

## Metabolic niche model

We begin by considering a model in which NFDS arises from within-host competition being stronger when strains are similar. Here we frame this as arising from metabolic niche overlap, although other effects such as strain-specific immunity could also give rise to the same model structure.

**Model description.** Again, we consider a two-locus, two-allele model (*G*={*ab*, *Ab*, *aB*, *AB*}). The full model structure is given in the Methods and the one locus case is represented in Fig 1 (see Fig A in S1 File Supplementary Information for the two-locus case). Briefly, hosts can be uncolonised (*S*), colonised with a single strain (*I•*) or co-colonised with two

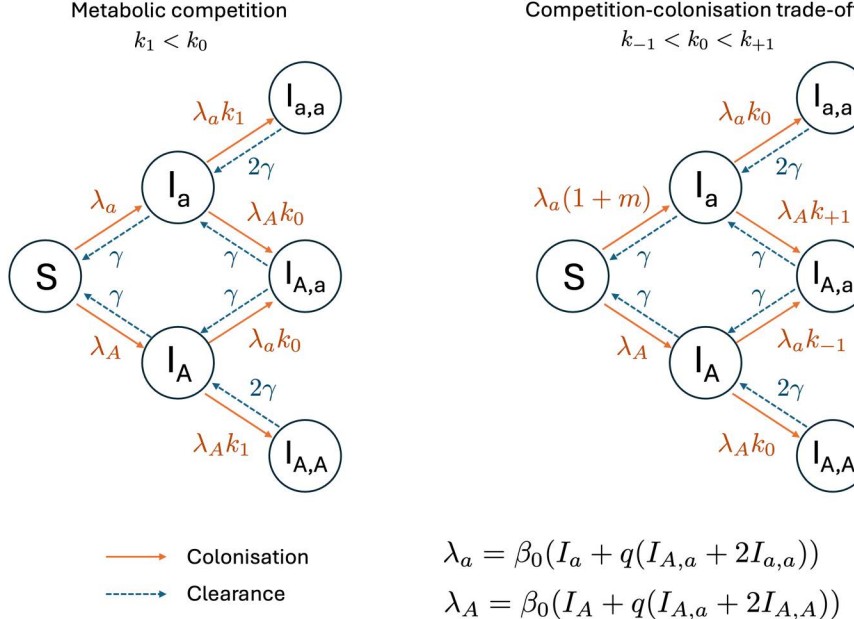

$$\lambda_a = \beta_0(I_a + q(I_{A,a} + 2I_{a,a}))$$
$$\lambda_A = \beta_0(I_A + q(I_{A,a} + 2I_{A,A}))$$

**Fig 1. NFDS models of metabolic niche or competition-colonisation trade-offs.** We represent the two epidemiological models for the case of one bi-allelic locus. In the metabolic niche model (left), co-colonisation rates are reduced when the resident and incoming strains share the same allele, modeled by parameters $k_1 < k_0$. In the competition-colonisation model (right), allele $a$ is "colonising", leading to an increased rate of primary colonisation $m$, while the competitive allele $A$ leads to higher rates of co-colonisation on average. Co-colonisation rates are dependent on the difference in number of competitive alleles between the incoming and the resident strain $\Delta$ through the parameters $k_\Delta$. $\lambda_a$ and $\lambda_A$ represent respectively the partial forces of infection of allele $a$ and $A$, taking into account that strains in co-colonisation have reduced transmission rates through parameter $q$. Solid orange lines represent colonisation and dashed blue lines represent clearance.

strains ($I_{\bullet,\bullet}$). Colonisation is cleared at rate $\gamma$, hosts become colonised through density-dependent transmission at baseline rate $\beta_0$. Onward transmission of each strain from a co-colonised host is scaled down by a factor $q$ compared to singly colonised hosts, reflecting reduced density within the host. A resident strain inhibits co-colonisation by incoming strains, modifying the transmission rate to already colonised hosts by a competition efficiency parameter $k$:

- $k_0$ If the incoming strain differs from the resident strain at both loci (i.e., no shared alleles)

- $k_1$ If the incoming strain shares one allele with the resident strain

- $k_2$ If the incoming strain shares both alleles with the resident strain

These parameters represent metabolic niches wherein different alleles provide access to different resources, with $k_0 > k_1 > k_2$. Strains sharing metabolic pathways (i.e., with the same allele at a locus) have a high niche overlap, competing for the same resources within the host, while strains with different metabolic capabilities (i.e., with different alleles) can coexist within a host more readily. This gives rise to NFDS at each locus, driving allele frequencies to 0.5 without the need for an explicit equilibrium frequency like was used in the phenomenological model (1) (see Supplementary Note A and Fig C in S1 File Supplementary Information for an analysis of the NFDS in the one-locus case). We explore in Fig B in S1 File Supplementary Information the neutral case ($k_0 = k_1 = k_2 = 1$) to show that the model structure itself does not influence the gene frequencies and LD dynamics.

**Metabolic niche differentiation leads to frequency-dependent effects on allele associations.** To understand the dynamics of LD in this model, we derive the expression for epistasis $w_{AB}$ (Equation (3)). Assuming $q = 1$ for simplicity:

$$w_{AB} = \beta_0(I_{AB} + I_{ab} - I_{Ab} - I_{aB})(k_0 - 2k_1 + k_2) \tag{5}$$

This expression indicates that epistasis is dependent on both the strain composition of the population and parameters relating to metabolic competition. The strain density term can be decomposed into the sum of the density of strains associated with positive LD ($AB$, $ab$), minus the sum of the density of strains associated with negative LD ($Ab$, $aB$). Thus, in this model, epistasis acts to reinforce or abolish existing LD, depending on the sign of the competition term.

The competition term $(k_0 - 2k_1 + k_2)$ reflects how NFDS combines across loci:

**If $k_1 = (k_0 + k_2)/2$**, NFDS combines additively across loci. Compared to co-colonising a host carrying an identical strain (i.e., two shared alleles, competition efficiency $k_2$), each additional divergent allele provides the same additional benefit. In this case, epistasis is zero and there is no effect on strain structure.

**If $k_1 < (k_0 + k_2)/2$**, NFDS combines super-additively (Fig 2a, red line). Compared to co-colonising a host carrying an identical strain ($k_2$), one divergent allele ($k1$) provides a small benefit relative to two divergent alleles ($k_0$). The biological interpretation here is that both loci are important: one private resource ($k_1$) facilitates co-colonisation only to a limited extent compared to both resources being private ($k_0$). The $(k_0 - 2k_1 + k_2)$ term is positive: epistasis will therefore act to reinforce existing LD. This produces bistability (Fig 2b and Fig D in S1 File Supplementary Information): LD reaches either $D' = \pm 1$ at equilibrium without recombination, depending on the initial sign of $D$.

**If $k_1 > (k_0 + k_2)/2$**, NFDS combines sub-additively (Fig 2a, grey line). Compared to co-colonising a host carrying an identical strain ($k_2$), one divergent allele ($k1$) provides a large benefit relative to two divergent alleles ($k_0$). The biological interpretation here is that there is redundancy between the loci: one private resource ($k_1$) facilitates co-colonisation to a large extent, and the benefit of the other additional resource is limited ($k_0$). The $(k_0 - 2k_1 + k_2)$ term is negative: epistasis will therefore act to abolish existing LD, driving $D'$ to zero at equilibrium (Figs 2b and D in S1 File Supplementary Information).

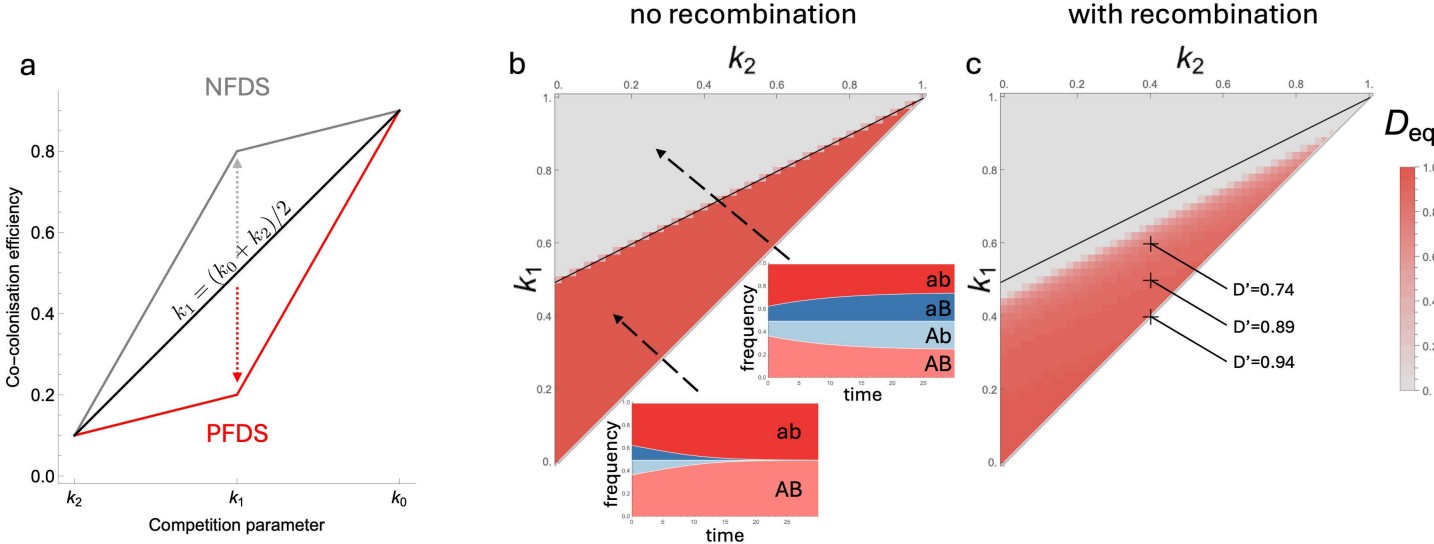

**Fig 2. Different competition regimes lead to maintenance or abolishment of strain structure.** In **(a)**, we show how different values of the competition efficiencies $k_0$, $k_1$ and $k_2$ influence epistasis and therefore strain structure according to Eq (5). Equilibrium $D'$ (t = 5000) is shown as a function of the competition parameters $k_1$ and $k_2$ in the absence **(b)** and the presence **(c)** of recombination. Müller plots in panel (b) show the dynamics of the cumulative frequencies of all genotypes through time, depending on the competition parameters, starting for an initial LD of $D' = 0.4$. Some specific values of $D'$ are noted on panel **(c)**. Parameter values used: $\beta_0 = 2$, $b = 4$, $\gamma = 2$, $d = 1$, $k_0 = 1$, $q = 0.5$, **(b)** $\sigma = 0$ and **(c)** $\sigma = 0.02$. Initial allele frequencies were initialised to $p_A = p_B = 0.5$, and this was applied to every figure of this work unless stated otherwise.

In other words, NFDS combining super-additively across loci gives rise to positive frequency dependent selection (PFDS) on pairs of non-overlapping genotypes (i.e., *AB* and *ab*, or *Ab* and *aB*). This reinforces existing strain structure. On the other hand, NFDS combining sub-additively across loci gives rise to NFDS on genotypes. This abolishes strain structure, even in the absence of recombination. The model is only neutral with respect to LD when NFDS combines precisely additively. These results are robust to relaxing the assumption $q = 1$ as long as $q$ does not become too small, a regime where co-colonised hosts are considerably less infectious overall than singly colonised hosts ($q = 0.1$ for the parameter set we use, see Fig E in S1 File Supplementary Information). The behaviour is qualitatively unchanged for this metabolic competition model when we introduce asymmetry with one allele (*a* and *b*) at each locus conferring an additional benefit to transmission. The only exception is when this leads to the absence of the least fit genotype *AB* (see Supplementary Note B and Fig H in S1 File Supplementary Information). We also explore the effect of asymmetric recombination which could play an important role in maintaining strain structure [20]. We find that asymmetric recombination is slightly less destructive of LD than symmetric recombination (see Fig I in S1 File Supplementary Information).

Introducing recombination into the model (see Methods) accelerates the decay of LD when genotypes are under NFDS (Fig F in S1 File Supplementary Information). When genotypes are under PFDS, the addition of recombination has a two-fold effect (Fig 2). Firstly, the parameter space in which linkage disequilibrium is promoted decreases: in the presence of recombination, existing allele associations are abolished unless PFDS is strong enough to overcome its effects. Secondly, in the parameter space in which strain structure is maintained, $D' < 1$ at equilibrium due to the balance between the effects of epistasis and recombination. These results are robust to deviations in initial allele frequencies (see Fig G in S1 File Supplementary Information), except for specific cases: (i) absence of epistasis without recombination where only the first terms of Equation 4 drive LD dynamics and (ii) PFDS with initial LD negative but close to zero, where the first terms of equation 4 can push LD towards a positive value early in the dynamics, and this positive LD is then reinforced by epistasis.

In summary, the metabolic niche model demonstrates that epistatic interactions resulting from NFDS processes combining across loci can lead either to abolishment or maintenance of allele associations, depending on the geometry of the generated fitness effects.

## Competition-colonisation trade-off model

We now explore a second epidemiological model, in which NFDS arises through a different mechanism: a competition-colonisation trade-off. This type of NFDS arises when one allele provides an advantage in transmitting to uncolonised hosts ('colonisation') while the other leads to better rates of co-colonisation to already colonised hosts ('competition'). This gives rise to NFDS because more colonisers decreases the ratio of uncolonised hosts to single infected hosts thus favouring competitors, and *vice versa*. This type of trade-off can arise from bacteriocin systems [29] or metabolic genes adapted to high vs low-nutrient (i.e., already colonised) environment.

**Model description** We use the same co-colonisation SIS framework, which is described in the methods, and the one locus case is represented in Fig 1 (see Fig A in S1 File Supplementary Information for the two-locus case). In this model, each locus has two alleles: a 'colonising' allele (*a* and *b*), which increases transmission to uncolonised hosts (*S*), and a 'competitive' allele (*A* and *B*), which increases transmission to already colonised hosts. We assume additive effects of the colonising alleles such that the primary colonisation rates of the *i* genotype, $\beta(i, S)$ is:

$$\beta(ab, S) = \beta_0(1 + 2m)$$
$$\beta(Ab, S) = \beta(aB, S) = \beta_0(1 + m)$$
$$\beta(AB, S) = \beta_0 \tag{6}$$

where ($\beta_0$) is the baseline transmission rate and $m$ is the benefit associated with the colonising allele. Transmission to already colonised hosts is modulated by $k_\Delta$, with $\Delta$ the difference in the number of competitive alleles between the incoming (*i*) and the resident (*j*) strains:

$$\beta(i,j) = \beta_0 k_{\Delta}(i,j) \tag{7}$$

where $\Delta \in \{-2, -1, 0, 1, 2\}$. For example $k_{\Delta}(Ab, AB) = k_{-1}$ and $k_{\Delta}(AB, ab) = k_{+2}$, with $0 \leq k_{-2} < k_{-1} < k_0 < k_{+1} < k_{+2} \leq 1$. Note that unlike the metabolic niche model, the alleles are not symmetric. However, genotypes $Ab$ and $aB$ are now phenotypically equivalent. This mechanistic model, similarly to the metabolic niche model, also gives rise to an equilibrium frequency favoured by selection, without it being an explicit component. We explore this model and the resulting NFDS and equilibrium frequency in the case of a single bi-allelic locus in Supplementary Note C and Fig J in S1 File Supplementary Information.

**Competition-colonisation trade-offs lead to a variety of linkage disequilibrium patterns.** The epistatic term $w_{AB}$ for $q = 1$ is given by (see Methods):

$$w_{AB} = \beta_0 \left( I_{ab}(k_{+2} - 2k_{+1} + k_0) + I_{AB}(k_{-2} - 2k_{-1} + k_0) + (I_{aB} + I_{Ab})(k_{+1} - 2k_0 + k_{-1}) \right) \tag{8}$$

As before, this expression can be decomposed into $k$-parameter terms multiplying the densities of strains. For each $I_i$ term, the corresponding $k$-parameter term captures information about how effectively genotypes can co-colonise hosts already carrying strain $i$. More specifically, each term reflects the impact of additional competitive alleles on the co-colonisation efficiency. For example, the first term $(k_{+2} - 2k_{+1} + k_0)$ represents the difference between i) the incoming strain having two vs one competitive allele $(k_{+2} - k_{+1})$ and ii) the incoming strain having one vs no competitive alleles $(k_{+1} - k_0)$. Thus, as before, these terms capture how effects combine across loci. Again, this can be represented by the geometry of the parameter curve (Fig 3a): negative terms correspond to the sub-additive effects (the second competitive allele has a smaller effect than the first) and a concave curve; positive terms correspond to super-additive effects (the second competitive allele has a greater effect than the first) and a convex curve.

The consequences of how these effects combine across loci are more complex than in the metabolic niche model. If all terms are positive (i.e., the entire curve is convex), $w_{AB}$ is also always positive: competitive alleles are always most beneficial when on the same genome as their effect is synergistic, leading to positive LD (i.e., more $ab$ and $AB$ strains) (Fig 3b). Conversely, if all terms are negative (i.e., the entire curve is concave), $w_{AB}$ is also always negative: the effect of gaining additional competitive alleles is saturating as every additional allele provides a smaller benefit than the last, leading to negative LD (i.e., more $Ab$ and $aB$ strains) (Fig 3b).

The model gives rise to frequency-dependent epistasis when the curve of the competition parameters $k_{\Delta}$ is not strictly concave or convex (Fig 3a), or in other words when the $(k_{i+2} - 2k_{i+1} + k_i)$ terms in (8) are not all of the same sign. In these scenarios, an additional competitive allele can have either a synergistic or an antagonistic effect, depending on how many competitive alleles are already present. In this case, the overall effect on $w_{AB}$ depends on strain frequencies. For example, if the terms multiplying $I_{ab}$ and $I_{AB}$ (i.e., strains associated with positive LD) are positive and the term multiplying the $I_{aB}$ and $I_{Ab}$ (i.e., strains associated with negative LD) is negative, $w_{AB}$ will have the same sign as the current LD. In other words, the model will give rise to PFDS on genotypes, reinforcing existing LD (Fig 3c). Conversely, if the terms multiplying $I_{ab}$ and $I_{AB}$ are negative and the term multiplying $I_{aB}$ and $I_{Ab}$ positive, $w_{AB}$ will have the opposite sign to current LD, therefore abolishing allele associations and driving absolute LD down (Fig 3b). Note that the equilibrium D' for the negative frequency-dependent epistasis case is not necessarily 0, and depends on the relative frequencies of $AB$ and $ab$ genotypes, which depends for instance on the value of $m$.

To explore a diversity of possible geometries, we illustrate in Fig K in S1 File Supplementary Information and L in S1 File Supplementary Information the equilibrium LD for a variety of parameter sets, as described in Supplementary Note D in S1 File Supplementary Information. These figures show the robustness of the results for $q = 1$ and $q = 0.5$. Note however that we adjust the value of the colonising benefit $m$ to maintain an equilibrium frequency of 0.5. This is because a low value of $q$ disproportionately affects the competitive alleles, which are more likely to be found in co-colonising strains.

PLOS Computational Biology

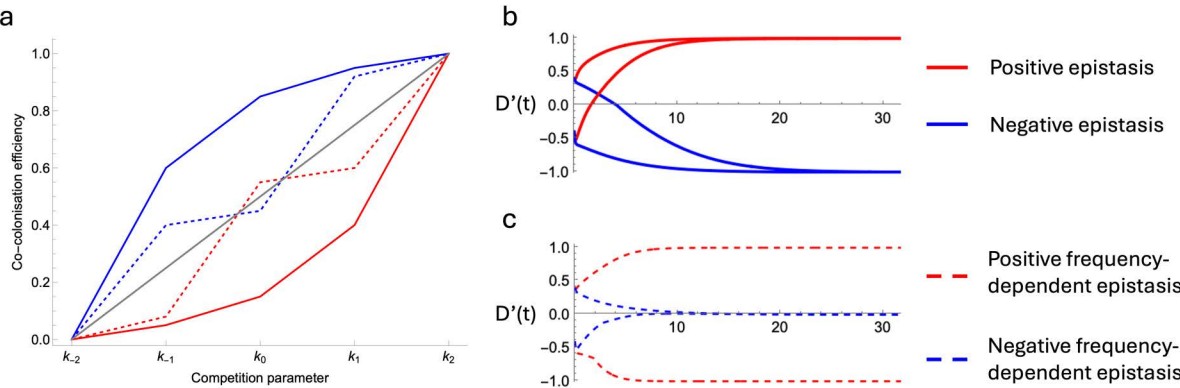

**Fig 3. The geometry of competitive interactions shapes epistasis and allelic associations in a competition-colonisation trade-off model.**
**(a)** A linear relationship between all the competition parameters $k_\Delta$ leads to additivity in how fitness effects combine across loci and thus no epistasis, see Eq (8). However a concave or a convex geometry will lead to epistasis of constant sign, as shown with solid lines. More complex geometries which are not purely concave or convex can produce epistasis which sign is frequency-dependent. The red dashed line scenario leads to PFDS wherein the most frequent pair of non-overlapping strains (those associated with either positive or negative linkage) is favoured. In contrast, the blue dashed line geometry leads to NFDS at the genotype level, favouring the least frequent pair of strains and thus abolishing allelic associations. D' trajectories are shown in (b,c) using the parameter sets shown in panel **(a)**, starting from both positive and negative initial D'. Other parameter values: $\beta_0 = 2$, $b = 4$, $\gamma = 2$, $d = 1$, $q = 1$, $m = 1.3$, $\sigma = 0$. See Fig K in S1 File Supplementary Information and L in S1 File Supplementary Information for equilibrium LD from simulations for a wider range of parameters.

Overall, depending on how the effects of individual alleles combine across loci, competition-colonisation trade-offs can generate i) constant epistasis generating and maintaining structure, ii) PFDS on pairs of non-overlapping genotypes reinforcing pre-existing structure or iii) NFDS at the genotype level, abolishing structure.

## Evidence for frequency-dependent effects on allele associations in *S. pneumoniae*

The key prediction from our models is that NFDS acting across multiple loci can give rise to frequency-dependent effects on allele associations. This prediction is difficult to test because, depending on parameter values, these effects can act to either abolish or reinforce allele associations. Furthermore, LD may also arise through other constant epistatic interactions between genes, or stochastic effects such as drift and hitch-hiking. Simply observing LD is thus not enough to discriminate between the mechanism suggested by our modeling and other possible causes. However, constant epistasis and PFDS make different predictions about LD *across* populations. Under constant epistasis, the same association is always favoured and the sign of LD must thus be the same across populations. PFDS, however, acts to reinforce existing structure; the sign of LD at equilibrium therefore depends on initial conditions. Pairs of genes with strong LD of different signs across different populations would therefore be consistent with PFDS, but not other forms of epistasis.

To test this idea, we examine patterns of linkage between 900 intermediate frequency genes across over 3000 pneumococcal genomes from three different locations: Thailand, Massachusetts and Southampton [30–32] represented in Fig M in S1 File Supplementary Information. To limit the impact of physical linkage on the chromosome, we apply a filter to only keep pairs of genes that are separated by at least 100kb on the genome. Note that this filtering is likely to remove some pairs of genes in strong positive epistasis (e.g., genes within the same operon).

We first analyse the functional categorisation of pairs of genes in LD (defined as $|D'| > 0.6$). Without the distance-based filtering, we find that pairs of genes sharing a function are more likely to be in LD than random gene pairs (Fig R in S1 File Supplementary Information). This effect is strongly decreased in the filtered dataset (Fig 4a), presumably due to the removal of genes for which proximity is an indicator of functional association (e.g., operons). Nevertheless, a significant effect remains where genes with certain functions are more likely to be strongly linked or unlinked to each other: cell cycle

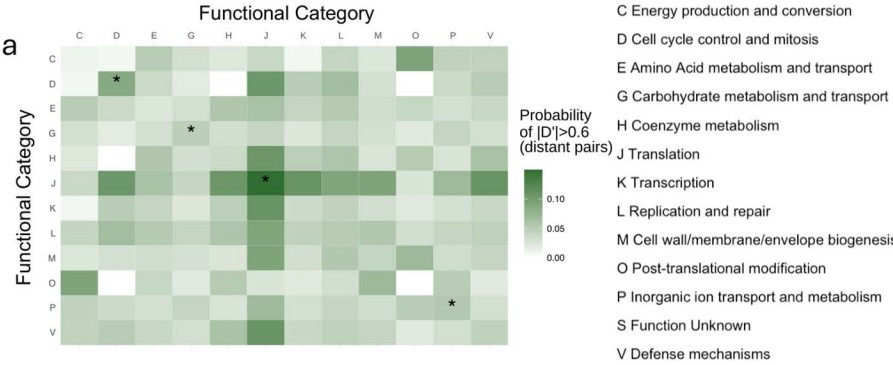

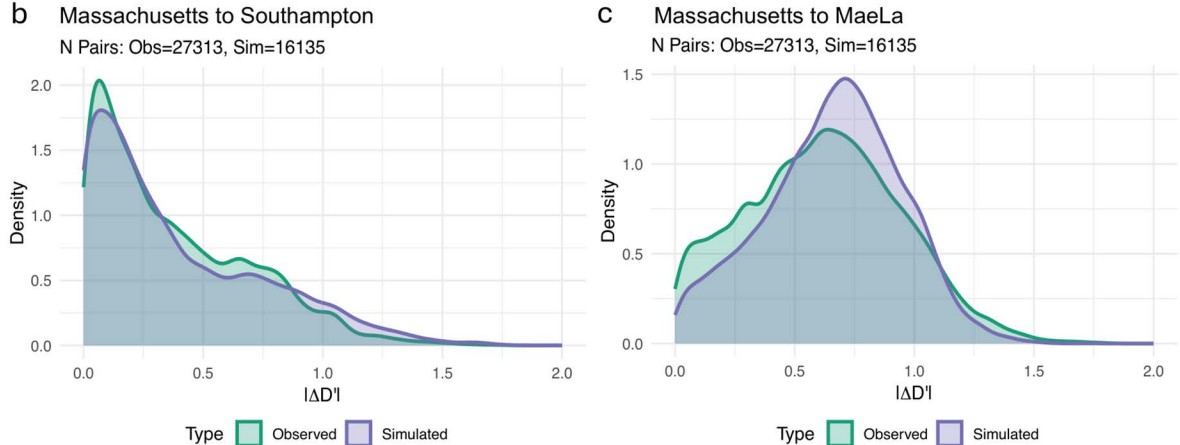

**Fig 4. Observed patterns of linkage disequilibrium in several populations.** (a) shows, according to the KEGG annotation of a focal gene, the probability to be in strong linkage disequilibrium (|D'|>0.6) with any given gene. An asterisk on the diagonal signifies that there is significantly more linkage for that combination (two genes with a shared KEGG category) compared to the rest of the row/column (one gene of that category with genes of different categories). See Fig R in S1 File Supplementary Information for the extension of this figure to all genes, without the 100kb distance filtering. In **(b,c)**, we study the distributions of absolute difference of LD across populations. We show for gene pairs highly linked in the Massachusetts dataset (|D'|>0.6) the distribution of change in linkage compared to (b) the Southampton dataset or (c) the MaeLa dataset. The observed changes distributions are shown in green and the purple distributions show the changes in the simulated population. Note how there are fewer pairs of genes being shown for the simulated populations, consistent with the observation that there is overall less LD in the simulated than the real dataset (Fig P in S1 File Supplementary Information). See Fig S in S1 File Supplementary Information for all the population pairwise comparisons.

control, carbohydrate metabolism and transport, translation and inorganic ion transport and metabolism. Genes in the translation category "J" (which notably includes integrases and transposases) show the strongest overall linkage, potentially reflecting the role of mobile genetic elements in LD distribution.

We then examine LD patterns across multiple populations, assessing the extent to which LD is conserved (strong LD of the same sign) or reversed (strong LD of opposite sign) across populations. For each pair of populations, we identify the set of gene pairs showing strong linkage (|D'|>0.6) in a focal population and look at their LD in the second population (Figs 4b, 4c and S in S1 File Supplementary Information). To distinguish between patterns arising from epistatic selection and those resulting solely from clonal population structure, we compare observed LD patterns against a phylogenetically constrained null model. In this null model, we simulate the evolution of accessory genes in both populations along the inferred core genome phylogenies using transition rates calibrated to match observed gene mobility (see Methods). This generates a "neutral" dataset that preserves the phylogenetic structure of the populations but is devoid of specific epistatic

interactions. We then compare the frequency of LD Conservation (maintaining |$D$'|$>$0.6 with the same sign) and LD Reversal (switching to |$D$'|$>$0.6 with the opposite sign) in the observed vs simulated data.

We find clear evidence of LD reversal when comparing the MaeLa population with the two other populations (Table 3): reversal is over twice as common as expected by chance (odds ratio (OR) >2 and $p<0.001$, two-sided empirical test) for all comparisons. This excess of reversals cannot be explained by clonal structure alone and is consistent with frequency-dependent epistasis reinforcing local LD. In contrast, we see no significant deviation from the null model when comparing the two Western populations (Massachusetts and Southampton). This likely reflects greater mixing between these populations (Fig M in S1 File Supplementary Information).

The evidence for LD conservation is not consistent across populations (Table 2): we find that observed levels of conservation are often indistinguishable from, or even lower than, those expected from the null model. This suggests an absence of constant-sign epistasis acting between long-range gene pairs. As we remove physically close gene pairs, the analysis does not reflect potential constant-sign epistasis between genes in close proximity (e.g., positive epistasis within the same operon). Thus, for genes separated by large genomic distances, we do not find evidence that classical constant sign-epistasis consistently maintains specific allele associations across different geographical regions.

In principle, LD maintained through PFDS on strain frequencies might not be robust to strong perturbations. To test this, we examined changes in LD following the introduction of the pneumococcal conjugate vaccine in Massachusetts (2001 vs. 2007). Observed conservation rates remained high ($\approx 34\%$) and were statistically similar to the null model ($OR \approx 1.0 - 1.1$). Similarly, there is no significant difference in LD reversal from the neutral expectation between the pre- and post-vaccine isolates. Therefore, although vaccination restructured the population and removed some key virulent strains, these results suggest that the LD distribution remained largely stable over this short timescale compared to the divergence accumulated between geographically separated populations.

## Discussion

In this work, we show that NFDS (negative frequency-dependent selection, i.e., when rare traits are selected for) acting across bacterial genomes [11–13] is expected to shape bacterial strain structure. Phenomenological models of multi-locus NFDS typically assume that the effects of NFDS combine additively across different loci [11,20]. Here,

**Table 2. Significance of LD Conservation across populations and time (Two-Sided Empirical Test).**

| Comparison | Focal Population | N Pairs Observed | Observed Prop. Cons. | Simulated Prop. Cons. | Odds Ratio | p-value |
|---|---|---|---|---|---|---|
| MaeLa ↔ South | MaeLa | 11388 | 0.2425 | 0.3198 | 0.76 | < 0.001 |
| | South | 27428 | 0.1007 | 0.0601 | **1.68** | < 0.001 |
| Mass ↔ MaeLa | Mass | 27313 | 0.1064 | 0.0746 | **1.43** | < 0.001 |
| | MaeLa | 11388 | 0.2551 | 0.2837 | 0.90 | 0.022 |
| Mass ↔ South | Mass | 27313 | 0.3913 | 0.4260 | 0.92 | 0.007 |
| | South | 27428 | 0.3896 | 0.3047 | **1.28** | < 0.001 |
| Pre- ↔ Post-vaccine | 2007 | 44450 | 0.3483 | 0.3704 | 0.94 | 0.100 |
| | 2011 | 45410 | 0.3409 | 0.3583 | 0.95 | 0.146 |

The table shows linkage analysis results between population pairs using a two-sided empirical test. Columns represent: compared populations, which focal population was analysed (Focal Population), number of strong linkage pairs analysed in the observed datasets (N Pairs Observed), observed and simulated proportions of conservation events (Prop. Cons.) and p-values. The simulated conservation events proportion is the average over 1000 simulations. Odds-ratios over 1 are shown in bold.

we develop two mechanistic models of NFDS – within-host niches and competition-colonisation trade-off – and show that there is no *a priori* reason to assume effects combine additively. Deviating from this assumption leads to epistatic effects. Under some parameter regimes, the competition-colonisation model gives rise to constant-sign epistatic effects, which predict stable associations between loci. More interestingly, both models also give rise to emergent frequency-dependent effects on strain structure. Negative frequency-dependence actively abolishes allele associations, while positive frequency-dependence acts to reinforce existing allele associations. The direction of associations reinforced by positive frequency-dependent epistasis can therefore be in either direction, and is determined by historical contingency.

We examine patterns of association between intermediate frequency genes in more than 3000 pneumococcal genomes, from three different geographical populations. We find that strong positive or negative LD is more common than expected between functionally related genes. However this effect is, as expected, highly diminished when filtering out pairs of physically close genes, as it filters out for instance genes within an operon, which likely share similar functions and are strongly linked. Overall, this analysis suggests that some of these associations are driven by epistatic rather than stochastic or demographic effects resulting from the clonality of bacterial species.

We then assess the fate of strongly linked gene pairs across geographical populations, comparing observed patterns of LD conservation (consistent with both constant-sign epistasis and frequency-dependent epistatic effects) and reversal (consistent with frequency-dependent epistatic effects) against a null model conserving the phylogenetic structure of the population. We do not see clear evidence of LD conservation. These results are not straightforward to interpret. Firstly, they are affected by the difference in the number of LD pairs in the two populations (see Supplementary Note E in S1 File Supplementary Information). Secondly, the removal of physically close gene pairs underestimates the extent of positive epistasis (Table A in S1 File Supplementary Information). On the other hand, we do see a clear signal of LD reversal. LD reversal is not observed between Southampton or Massachusetts. This is not unexpected, given the extent of mixing between the populations (Fig M in S1 File Supplementary Information). Overall, these observations are consistent with the presence of frequency-dependent effects reinforcing epistasis where local historical contingency determines the direction of the association.

Our empirical results support the role of frequency-dependent effects in shaping strain structure. We found evidence of LD reversal across populations, consistent with frequency-dependent interactions, but no clear signal of LD conservation beyond what is expected from the core phylogeny. This suggests that while constant-sign epistasis may occur at short ranges (such as within operons), it does not appear to be a force maintaining non-random associations between distant loci across the species. However, assessing the practical importance of these effects on observed LD patterns remains a challenge. Firstly, predicted frequency-dependent effects can act to either reinforce or abolish allele associations depending on how effects combine across loci. While our null model analysis supports that the observed patterns of LD reversal cannot be explained by clonality alone, disentangling exactly what part of the observed effect is due to frequency-dependent epistasis remains difficult. Empirical estimates of variation in colonisation efficiency are feasible (e.g., [33]) and could help to delineate the contribution of epistatic effects. However, estimating how these effects are influenced by multiple loci requires large sequenced surveys of bacterial colonisation.

Secondly, NFDS is hypothesised to maintain intermediate frequencies for genes across a wide range of functions, including immunity, metabolism, direct competition and antibiotic resistance [11]. We hypothesise that the emergence of frequency-dependent epistasis from multi-locus NFDS is a general effect. This work focuses on two mechanisms. The first is within-host niches, which we conceptualise as arising from metabolic differentiation, though it could also reflect immunity. The second is a competition-colonisation trade-off, which can arise from bacteriocin systems [29]. Previous work has considered the role of antigen-specific acquired immunity in maintaining allele associations [22,27], with similar frequency-dependent epistatic effects also arising in these models [27]. Whether similar effects are predicted to arise for other

functions is, in principle, easy to address through modeling. However, the mechanisms giving rise to NFDS across the genome are not well characterised.

Thirdly, in the mechanistic models we consider, both loci are under the same form of NFDS, and it is unclear whether these effects also arise between loci under *different* forms of NFDS. Previous modeling suggests that NFDS on metabolic and antigenic loci can lead to LD between the two types of alleles [34]. In the model giving rise to these results, NFDS on antigenic and metabolic loci is assumed to arise through relatively similar mechanisms (i.e., effects on colonisation efficiency). It is unclear whether the same applies if NFDS arises through very different mechanisms for different loci. Similarly, addressing how effects combine across more than two loci under NFDS is also important.

Fourth, we analyse a simple model focusing on a single pair of loci. Microbial genomes are composed of thousands of loci potentially interacting through epistasis. We cannot expect that the behaviour of a pair of loci is independent from the rest of the genome and so pairwise observations do not tell the whole story. For instance, it is expected that when considering many loci with weak epistasis and recombination, there can be an emergent regime where blocks of interacting genes can be locked and transmitted together [35]. Our work suggests that such epistatic interactions are plausible between many genes involved in within-host interactions.

Overall, our theoretical results represent qualitative insights into the role of NFDS in shaping strain structure and our empirical results suggest patterns of LD consistent with these effects. By using a simulated null model which conserves the phylogenetic structure of the species, we take into account the maintenance or reversal of linkage due to clonal expansion, which would otherwise be a major confounding effect. A number of other mechanisms such as demographic bottlenecks [6], asymmetric recombination [20], heterogeneity in recombination rates, and population structure [36]– also plausibly contribute to observed strain structure. Quantifying the relative impact of these processes will require more complex models than the ones presented in this work. While our simulation controls for the average rate of gene gain and loss along the tree, it assumes a homogeneous process across the genome. Future work could incorporate more complex null models that account for recombination hotspots, gene-specific mobility rates or physical proximity to further refine these estimates.

Our results have implications for understanding the response of pathogen populations to disturbances, such as vaccinations. Associations reinforced by positive frequency-dependent epistasis are contingent on the population's evolutionary history. Such associations may therefore not be stable to large perturbations, leading to potentially restructuring of patterns of associations. We tested this in the Massachusetts data following the introduction of the pneumococcal conjugate vaccine and observed large-scale conservation of LD and few cases of LD sign reversal. The observed rate of LD reversal was not significantly different from the null model, in contrast with the geographic comparisons (*OR* > 2.0). This suggests that while vaccination could have perturbed frequency-dependent epistatic interactions, the LD distribution remained largely unchanged over the short timescale observed (6 years). This is in contrast to the divergence seen between geographically separated populations. A caveat here is that this is a single example of vaccine introduction, and it would be interesting to study more examples of such longitudinal data. More broadly, models of NFDS are used to predict which strains will increase in frequency following removal of vaccine-targeted serotypes [11,13,37]. Such models make considerably better predictions than naive models (e.g., $R^2$ between observed and predicted post-vaccine strain frequencies are 0.2 vs 0.02 for a NFDS model vs a model based on pre-vaccine strain frequencies). Yet, accurately predicting post-vaccine strain frequencies remains very challenging. Accounting for epistatic interactions may improve the predictive power of such models.

The presence of stable strain structure requires both the presence of allelic diversity and epistatic mechanisms which promote non-random associations between alleles. Here, we show that NFDS, which gives rise to the former, can also impact the latter through frequency-dependent epistasis. NFDS is only neutral with respect to allele associations under very specific assumptions, and, for genes under a similar form of NFDS, there is no *a priori* reason to expect these

assumptions to hold. These mechanisms that maintain genetic diversity across loci are thus also fundamentally linked to how this diversity is structured.

## Methods

### Measures of linkage disequilibrium

Linkage disequilibrium can be measured with the D metric defined as:

$$D = p_{AB} - p_A p_B \tag{9}$$

which is bounded by $-0.25 \leq D \leq 0.25$. However this interval is reduced as frequencies of alleles $A$ or $B$ diverge from 0.5. In order to get a more interpretable metric, we use throughout most of the figures in this article the metric $D'$ which is bounded by -1 and 1 independently of the frequencies of individual alleles (provided they are not 0 or 1):

$$D' = \begin{cases} \dfrac{D}{\min(p_A(1-p_B),\,(1-p_A)p_B)}, & \text{if } D > 0, \\ \dfrac{D}{\min(p_A p_B,\,(1-p_A)(1-p_B))}, & \text{if } D < 0. \end{cases} \tag{10}$$

### Multi-locus NFDS

In this section, we present how we implement recombination in the multi-locus NFDS model. We use mass-action recombination where two genotypes exchange an allele at a rate dictated by their respective densities and a recombination rate $\sigma$. This gives the following expression for the changes in density due to recombination $\varphi_{ij}$ of genotype $ij$:

$$\varphi_{ab} = \varphi_{AB} = \sigma\left(X_{aB}X_{Ab} - X_{AB}X_{ab}\right) \tag{11}$$

$$\varphi_{aB} = \varphi_{Ab} = \sigma\left(X_{AB}X_{ab} - X_{aB}X_{Ab}\right) \tag{12}$$

the two terms in each expression respectively showing the creation and the disappearance of the focal genotype through recombination.

### SIS with co-colonisation

We consider an SIS (susceptible-infectious-uncolonised) for commensal bacteria circulating in a homogeneous host population. Uncolonised hosts are characterised by a birth rate $b$ and a death rate $d$. There are several strains of commensals, and we denote with $I_i$ hosts colonised by strain $i$. Commensals are transmitted through a mass action mechanism with a baseline transmission rate of $\beta_0$, and can be cleared with a rate $\gamma$. There is no virulence (or additional mortality) associated with the carriage of a commensal. In this model we allow for co-colonisation by two commensals, which may or may not be of the same strain. We denote with $I_{i,j}$ such a host co-colonised by both strains $i$ and $j$. Note that there is no meaning to the order of strains in the notation and so $I_{i,j}$ and $I_{j,i}$ can be used interchangeably. Each co-colonising bacteria is transmitted at rate $\beta_0 q$ where $q$ is the efficiency associated with being in co-colonisation. This means that if $q=1$, the co-colonised host $I_{i,i}$ is exactly twice as infectious as host $I_i$. There is no interaction between co-colonising bacteria acting on the duration of carriage and so a co-colonised host $I_{i,j}$ leaves the co-colonised compartment at rate $2\gamma$: it becomes $I_i$ or $I_j$ at rate $\gamma$ respectively. Two strains in co-colonisation can recombine upon secondary colonisation with probability $\sigma$, leading to a term $\Delta_{rec}$ detailed in the next section. The resulting system of ordinary differential equations is:

$$\frac{dS}{dt} = b - Sd - S\underbrace{\left(\sum_{x\in G} I_x\beta(x,S) + q\sum_{x,y\in G} I_{x,y}(\beta(x,S)+\beta(y,S))\right)}_{\text{primary colonisation}} + \underbrace{\gamma\sum_{x\in G} I_x}_{\text{recovery}}$$

$$\frac{dI_i}{dt} = \underbrace{SI_i\beta(i,S) + Sq\left(2I_{i,i}\beta(i,S) + (1-\sigma)\sum_x I_{i,x}\beta(i,S)\right)}_{\text{primary colonisation}} + \underbrace{\gamma\left(I_{i,i} + \sum_{x\in G} I_{i,x}\right)}_{\text{recovery}}$$

$$- I_i\underbrace{\left(\sum_{x\in G} I_x\beta(i,x) + q(1-\sigma)\sum_{x,y\in G} I_{x,y}\big(\beta(x,i)+\beta(y,i)\big)\right)}_{\text{co-colonisation}} - \underbrace{I_i(d+\gamma)}_{\text{death, recovery}} + \Delta_{rec}(I_i)$$

$$\frac{dI_{i,j}}{dt} = \underbrace{I_iI_j(\beta(i,j) + (1-\delta(i=j))\beta(j,i)) + q(1-\sigma)\left(I_i\left(\sum_{x\in G} I_{x,j}\beta(j,i)\right) + I_j\left(\sum_{x\in G} I_{i,x}\beta(i,j)\right)\right)}_{\text{co-colonisation}}$$

$$- \underbrace{I_{i,j}(d+2\gamma)}_{\text{death, recovery}} + \Delta_{rec}(I_{i,j})$$

$$(13)$$

## Recombination

As we model co-colonisation, we can model recombination directly at the level of interacting strains and not use a population-wide recombination rate. We model recombination as happening upon the colonisation of a new host, from a co-colonised host. From a co-colonised host $I_{i,j}$, if strain $i$ recombines with strain $j$ upon infection, strain $i$ will receive the allele of strain $j$ in a random locus. $\delta_z(i,j)$ is the probability, upon recombination, that genotype $i$ recombines with genotype $j$ to form genotype $z$. For instance:

$$\begin{cases} \delta_{aB}(ab, AB) & = 1/2 \\ \delta_{aB}(aB, AB) & = 1/2 \\ \delta_{aB}(ab, Ab) & = 0 \\ \delta_{AB}(AB, AB) & = 1 \end{cases}$$

$$(14)$$

The resulting terms of growth rate dependent on recombination are then given by:

$$\Delta_{rec}(I_i) = q\sigma(S\underbrace{\sum_{x,y\in G} I_{x,y}\beta(i,S)\big(\delta_i(x,y)+\delta_i(y,x)\big)}_{\text{primary colonisation}} - I_i\underbrace{\sum_{x,y,j\in G} I_{x,y}\big(\delta_j(x,y)\beta(j,i)+\delta_j(y,x)\beta(j,i)\big)}_{\text{co-colonisation}})$$

$$\Delta_{rec}(I_{i,j}) = q\sigma\bigg(\underbrace{\sum_{x,y\in G} I_{x,y}\Big(I_i\big(\delta_j(x,y)\beta(j,i)+\delta_j(y,x)\beta(j,i)\big) + I_j\big(\delta_i(x,y)\beta(i,j)+\delta_i(y,x)\beta(i,j)\big)\Big)}_{\text{co-colonisation}}\bigg)$$

$$(15)$$

## Evolutionary dynamics of LD

Evolutionary dynamics can be monitored through the dynamics of the frequency of each allele and of linkage disequilibrium [28].

$$\frac{dp_A}{dt} = w_A p_A (1 - p_A) + w_B D + w_{AB}(1 - p_A)p_{AB}$$

$$\frac{dp_B}{dt} = w_B p_B (1 - p_B) + w_A D + w_{AB}(1 - p_B)p_{AB}$$

$$\frac{dD}{dt} = w_A(1 - 2p_A)D + w_B(1 - 2p_B)D + w_{AB}(p_A p_B + D)((1 - p_A)(1 - p_B) - D)$$

(16)

The parameter $q$ is important in the expression of the epistasis $w_{AB}$ and dictates the effective force of infection by a certain strain depending on the number of hosts where this strain is in single infection and the number of hosts where that strain is co-colonising. We define the effective colonising density for strain $i$ $\varphi_i(t)$ as:

$$\varphi_i(t) = I_i(t) + q(I_{i,i}(t) + I_{i,ab}(t) + I_{i,Ab}(t) + I_{i,aB}(t) + I_{i,AB}(t))$$

(17)

and the per-capita effective rate of new infections by strain $ab$:

$$E_i(t) = \beta(i, S)S + \beta(i, ab)I_{ab}(t) + \beta(i, aB)I_{aB}(t) + \beta(i, Ab)I_{Ab}(t) + \beta(i, AB)I_{AB}(t)$$

(18)

Hence $E_i(t)\varphi_i(t)$ is the force of infection, or rate of new colonisation by the strain $i$. With these terms we can express the growth rate of strain $i$:

$$r_i(t) = \frac{N'_i}{N_i} = \frac{I'_i + I'_{i,i} + \sum_{j \in G} I'_{i,j}}{I_i + I_{i,i} + \sum_{j \in G} I_{i,j}}$$

$$= \beta_0 \frac{\varphi_i(t)E_i(t)}{N_i(t)} - (d + \gamma)$$

(19)

To finally get an expression for the epistasis:

$$w_{AB} = r_{AB} + r_{ab} - r_{aB} - r_{Ab}$$

$$= \frac{E_{AB}(t)\varphi_{AB}(t)}{N_{AB}(t)} + \frac{E_{ab}(t)\varphi_{ab}(t)}{N_{ab}(t)} - \frac{E_{aB}(t)\varphi_{aB}(t)}{N_{aB}(t)} - \frac{E_{Ab}(t)\varphi_{Ab}(t)}{N_{Ab}(t)}$$

(20)

In the special case of $q = 1$, a strain in co-colonisation is as infectious as a strain in a single infected host. This leads to the equality between the effective density and the total density of strain $i$ $\varphi_i(t) = N_i(t)$.

### Genome data

Genomes of *S. pneumoniae* were downloaded from NCBI BioProjects (MaeLa [30]:) PRJEB2357, PRJEB2393, PRJEB2395, PRJEB2479, PRJEB2480, (Southampton [31]:)PRJEB2417, (Massachusetts [32]:)PRJEB2632. The final dataset contains 2120 isolates from MaeLa, 605 from Massachusetts and 456 from Southampton. The phylogenetic tree for these isolates was downloaded on https://microreact.org/project/multilocusNFDS [11]. Gene sequences were clustered into clusters of orthologous genes using the easy-cluster workflow of MMSEQS2 [38]. Ancestral reconstruction of accessory gene content was carried out with PAST-ML using the JOINT prediction method [39] and protein annotation into KEGG functional categories was done with eggnog mapper v2 [40] using the emapper workflow and the diamond database.

## Gene filtering

All subsequent analysis of the genome data was carried out using R 4.3.1 [41] and the ape package [42]. Accessory genes were filtered with frequency threshold $0.1 < f < 0.9$ in each geographical population. Next, we only keep pairs separated by at least 100 kb on the chromosome. For each gene pair, we counted the frequency, among genomes where both genes are present, of the two genes being more than 100 kb apart or on different contigs. We kept only the pairs of genes for which this condition was true in more than 99% of genomes. We use this frequency based threshold because accessory genes are not always located in the same position.

## Phylogenetic null model simulation

To distinguish between linkage disequilibrium arising from epistatic selection and that resulting from clonal population structure, we generated a null model dataset by simulating gene gain and loss dynamics along the phylogenetic tree. Evolutionary simulations were performed using a continuous-time Markov chain (Mk model) implemented in the castor R package [43].

The goal is for each gene to find the gain and loss rates, and this can be decomposed into two steps: finding the relative gain and loss rates that correctly simulate the gene frequencies, and then scaling these rates to match the observed number of changes.

For each accessory gene $i$, the transition rates for gene gain ($q_{0 \to 1}$) and gene loss ($q_{1 \to 0}$) were defined to ensure that the simulated genes had the same distribution of frequencies ($f_i$) as found in the observed data:

$$q_{0 \to 1} = f_i \mu_i \tag{21}$$

$$q_{1 \to 0} = (1 - f_i) \mu_i \tag{22}$$

where $\mu_i$ is the baseline substitution rate for gene $i$ derived from the density of observed events of gains and losses ($E_i$) over the total tree length ($L$). This baseline was adjusted by a global scaling factor ($F$):

$$\mu_i = S \frac{E_i}{L} \tag{23}$$

The scaling factor was empirically calibrated to $F = 5.7$, a value that aligned the median distribution of total evolutionary changes in the simulated dataset with the median number of changes observed in the real dataset (see Fig O in S1 File Supplementary Information). This scaling factor is necessary and necessarily positive. We assess the parsimony score (number of gains and losses) in the observed dataset and we want to match this with the simulated dataset. To do so, we estimate $\mu_i$ so that each simulated gene $i$ changes the observed number of times. However, because in reality several gain and loss events can happen in the same branch, the estimated parsimony score computed from the tips of the tree is necessarily lower than the true number of changes that happened. Therefore the scaling factor is necessary, must be above one, and depends on the specific phylogenetic tree. The state of the root node for each gene was fixed to the state inferred from the ancestral reconstruction of the observed data.

Because the Mk model is stochastic, the simulated frequency of a gene may deviate from the observed frequency. To ensure the null model accurately reflected the observed gene frequencies, the simulation was re-run for every gene until its final frequency was within a $\pm 1\%$ margin of the observed frequency (see Fig N in S1 File Supplementary Information for the observed and simulated frequency distributions). We show in Fig P in S1 File Supplementary Information the distribution of $D'$ in the observed and simulated populations, and in Fig Q in S1 File Supplementary Information the

distribution of accessory genes on the phylogenetic tree in the observed and simulated Southampton population as an example. To robustly capture the variance expected under neutral evolution, we generated 100 independent replicate simulated populations.

**Strong linkage pairs across populations**

We calculated the *D'* measure of LD for all physically distant gene pairs to ensure statistical independence (pairs separated by distance thresholds described previously). For each pair of populations (e.g., $Pop_1$ vs. $Pop_2$), we identify "Strong Linkage" pairs, defined as having $|D'|>0.6$ in the reference population ($Pop_1$). We then observe the change in $|D'|$ for these specific pairs in the target population ($Pop_2$). Pairs were classified as Conserved if they maintained strong LD with the same sign ($|D'|_{pop2}>0.6$ and sign consistent) and reversed if they switched to strong LD with the opposite sign ($|D'|_{pop2}>0.6$ and sign flipped). We repeated this analysis in 1000 simulated replicate datasets, comparing LD changes between simulated populations, to obtain a null distribution for the mean rates of reversal and conservation events. Finally, we tested whether observed mean rates were significantly different from the neutral expectation. We obtain a p-value by comparing the observed mean rates to the empirical distribution. The results are found in Tables 2 and 3. An Odds Ratio (*OR* > 1) indicates that the event (Conservation or Reversal) occurs more frequently in the observed data than expected in the simulated populations where there is no epistasis.

To distinguish biological signal from phylogenetic inertia, we compared the observed rates of Conservation and Reversal against a null distribution derived from 1,000 simulated datasets generated under a neutral model. For each simulation replicate, we identified strongly linked pairs in the simulated focal population and calculated the resulting event rates in the simulated target population. Statistical significance was assessed using an empirical *P*-value, defined as the proportion of simulations where the simulated rate met or exceeded the observed rate ($P = \frac{N_{sim \geq obs}+1}{N_{total}+1}$). We quantified the effect size using an Odds Ratio (OR), calculated as the observed rate divided by the mean simulated rate. An *OR* > 1 indicates that the event (Conservation or Reversal) occurs more frequently in the observed data than expected under neutral phylogenetic drift, suggesting the action of selection. Results are detailed in Tables 2 and 3.

**Fate of strong linkage pairs**

We calculated *D'* for all physically distant gene pairs (pairs separated by distance thresholds described previously). For each directed comparison between two populations ($Pop_1 \rightarrow Pop_2$), we identified "Strongly Linked" pairs, defined as

**Table 3. Significance of LD Reversal across populations and time (Two-Sided Empirical Test).**

| Comparison | Focal Population | N Pairs Observed | Observed Prop. Rev. | Simulated Prop. Rev. | Odds Ratio | p-value |
|---|---|---|---|---|---|---|
| | | | | | | |
| MaeLa ↔ South | MaeLa | 11388 | 0.0587 | 0.0232 | **2.53** | < 0.001 |
| | South | 27428 | 0.0244 | 0.0044 | **5.60** | < 0.001 |
| Mass ↔ MaeLa | Mass | 27313 | 0.0113 | 0.0032 | **3.53** | < 0.001 |
| | MaeLa | 11388 | 0.0272 | 0.0122 | **2.22** | < 0.001 |
| Mass ↔ South | Mass | 27313 | 0.0059 | 0.0078 | 0.76 | 0.577 |
| | South | 27428 | 0.0059 | 0.0056 | **1.05** | 0.921 |
| Pre- ↔ Post-vaccine | 2007 | 44450 | 0.0393 | 0.0416 | 0.94 | 0.709 |
| | 2011 | 45410 | 0.0384 | 0.0402 | 0.96 | 0.752 |

The table shows linkage analysis results between population pairs using a two-sided empirical test. Columns represent: compared populations, which focal population was analysed (Focal Population), number of strong linkage pairs analysed in the observed datasets (N Pairs Observed), observed and simulated proportions of reversal events (Prop. Rev.) and p-values. The simulated reversal events proportion is the average over 1000 simulations. Odds-ratios over 1 are shown in bold.

having $|D'| > 0.6$ in the focal population ($Pop_1$). We then determined the status of these specific pairs in the target population ($Pop_2$). Pairs were classified as conserved if they maintained strong LD with the same sign ($|D'|_{Pop_2} > 0.6$; consistent sign) and reversed if they switched to strong LD with the opposite sign ($|D'|_{Pop_2} > 0.6$; flipped sign).

We then compared the observed rates of conservation and reversal against a null distribution derived from 1,000 simulated datasets generated under the neutral model. For each simulation replicate, we identified strongly linked pairs in the simulated focal population and calculated the resulting event rates in the simulated target population. Statistical significance was assessed using a **two-sided** empirical $P$-value. This was defined as the proportion of simulations where the absolute deviation of the simulated rate ($r_{sim}$) from the mean simulated rate ($\bar{r}_{sim}$) was greater than or equal to the absolute deviation of the observed rate ($r_{obs}$):

$$P = \frac{\sum_{i=1}^{N} \mathbb{I}(|r_{sim,i} - \bar{r}_{sim}| \geq |r_{obs} - \bar{r}_{sim}|) + 1}{N + 1} \tag{24}$$

where $N$ is the total number of simulations (1,000) and $\mathbb{I}$ is the indicator function. We quantified the effect size using an Odds Ratio (OR), calculated as the observed rate divided by the mean simulated rate. An $OR \neq 1$ accompanied by a significant $P$-value indicates that the event frequency differs significantly from neutral expectations, consistent with epistasis (e.g., $OR > 1$ suggests enrichment). Results are detailed in Tables 2 and 3.

## Supporting information

**S1 File. Supplementary information contains supplementary figures and supplementary notes referenced in the main text.**
(PDF)

## Author contributions

**Conceptualization:** Martin Guillemet, Sonja Lehtinen.

**Formal analysis:** Martin Guillemet.

**Investigation:** Martin Guillemet.

**Supervision:** Sonja Lehtinen.

**Writing – original draft:** Martin Guillemet.

**Writing – review & editing:** Martin Guillemet, Sonja Lehtinen.

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
