## [Decision Letter · Decision Letter 0]

3 Sep 2025

PCOMPBIOL-D-25-01407

Emergent epistasis mediates the role of negative frequency-dependent selection in bacterial strain structure

PLOS Computational Biology

Dear Dr. Guillemet,

Thank you for submitting your manuscript to PLOS Computational Biology. After careful consideration, we feel that it has merit but does not fully meet PLOS Computational Biology's publication criteria as it currently stands. Therefore, we invite you to submit a revised version of the manuscript that addresses the points raised during the review process.

Please submit your revised manuscript within 60 days Nov 03 2025 11:59PM. If you will need more time than this to complete your revisions, please reply to this message or contact the journal office at ploscompbiol@plos.org. Please include the following items when submitting your revised manuscript:

We look forward to receiving your revised manuscript.

Kind regards,

Tommy Tsan-Yuk Lam, Ph.D.

Academic Editor

PLOS Computational Biology

Jennifer Flegg

Section Editor

PLOS Computational Biology

**Journal Requirements:**

4) We notice that your supplementary Figures, Tables, and information are included in the manuscript file. Please remove them and upload them with the file type 'Supporting Information'. Please ensure that each Supporting Information file has a legend listed in the manuscript after the references list.

5)  Thank you for indicating that "The genome data used in this work is publicly available and the identifiers are described in the manuscript." Please update your Data Availability Statement in the online submission form to include the accession number(s) of the dataset(s).

6) Thank you for stating "No competing interests." Please revise your current Competing Interest statement to the standard "The authors have declared that no competing interests exist."

**Reviewers' comments:**

Reviewer's Responses to Questions

Reviewer #1: This manuscript by Guillemet & Lehtinen is an intriguing exploration of the interaction between epistasis and frequency-dependent selection on bacterial strain structure. This builds on previous work on McLeod et al (https://www.pnas.org/doi/10.1073/pnas.2401578121), which focused on immune selection. This manuscript extends the work to consider competition between commensals through two different mechanisms.

My major criticisms of the work are:

[1] The title and Discussion refer to “emergent epistasis”. An emergent property is one that the system has, but the components do not. Yet in each case, the epistasis is assumed by the equations that define the model, rather than being inferred from observations of simulation results.

To clarify, the Discussion states, “there is no a priori reason to assume effects combine additively. Deviating from this assumption leads to emergent epistatic effects”. Yet one of the original definitions of epistasis, by Fisher, is “any statistical deviation from the additive combination of two loci in their effects on a phenotype” (https://pmc.ncbi.nlm.nih.gov/articles/PMC2689140/#BX2). Therefore the epistasis is assumed, intrinsic to the equations, not emergent.

The results shown in Figure 3 are the closest to being emergent. However, in these simulations, the epistasis still requires that the effect of a two allele deviation is not equal to double the effect of a one allele deviation. This again is a violation of additivity, and therefore implicit epistasis within the equations that define the model.

The authors should therefore comment on how their work compares with previous analyses of the effect of epistasis on the emergence of strain structure in a recombining population (e.g. https://doi.org/10.1073/pnas.0812560106).

[2] The validation of the model predictions using genomic data do not appear to consider physical linkage through spatial proximity of genes in a chromosome. The null distribution to which the biological data are compared was generated through shuffling the presence of genes across genomes, which is inconsistent with the known genome biology of these bacteria (e.g. the structuring of genes into operons), and their population structure (which is not panmictic). Therefore is it unsurprising that the observed distribution deviates from the null distribution. The authors should limit their comparisons to genes separated by a significant distance (e.g. 100 kb) to minimise the effects of physical linkage.

The authors do sensibly limit their analysis to genes that are gained or lost at least 30 times on the tree. However, such rapidly moving genes are likely to be mobile genetic elements. These often represent blocks of many genes that, by definition, move as a single linked unit. Therefore it is to be expected that they will exhibit high LD. It is probable that these structures account for the lower-than-expected proportion of reversals observed following vaccine introduction. This explanation fits with the authors noting there are many integrases and transposes in the set of genes showing the highest linkage, “potentially reflecting the role of mobile genetic elements in LD”. This issue should also be fixed by using the distance threshold suggested above to only look at distantly-separated genes.

Additionally, if there are epistatic interactions between genes, why would selection not result in them being found in the same operon? At which point, would it not be expected that they would be co-inherited, and therefore cease to affect population structure?

[3] Equation 1 features “the allele frequency favored by selection” for each allele, which features in the fitness calculations. However, these do not appear to be mentioned again in the paper. Figure 2b suggests that all four favored frequencies are 0.5. Is this the case? Do all the equations generalize to any favored frequency? What effect do unequal frequencies have?

Additionally, it would be reassuring to see the framework used to undertake neutral simulations, with the results included in the SOM for comparison with the NFDS simulations.

My minor comments are:

[a] Equation (3) – the top equation only applies when fitneses are additive, according to McLeod et al (page 6, SOM) – this should be clarified

[b] Equation (4) – does not match equation (16), or McLeod et al SOM equation 22c – can the authors explain this discrepancy?

[c] L129 – sigma not defined here, only in Methods

[d] Figure 2 – why is there an infinity sign next to D’ in the scale? (see also Fig. S6) Also, the zero of beta_0 is not subscript in many figure legends.

[e] L256 – “D’egative”

[f] L312 – there is no Figure 4d.

[g] Equation 9 – "B" should be subscript

[h] L611-613: “bacteria” is plural

[i] Figure S3 – should be “null or negative”

[j] The graphs in panel a of Figure S7 and S8 appear to be misformatted.

[j] "Streptococcuc" in the keywords

Reviewer #2: Summary:

This manuscript explores how multi-locus negative frequency-dependent selection (NFDS) generates emergent epistasis, thereby shaping bacterial strain structure. Using two mechanistic epidemiological models (metabolic niche differentiation and competition–colonisation trade-offs), coupled with genomic analysis of >3000 Streptococcus pneumoniae genomes, the authors argue that frequency-dependent epistasis explains patterns of linkage disequilibrium (LD). The study aims to generalise beyond antigen-driven NFDS to broader mechanisms.

The study is conceptually rich, methodologically sophisticated, and relevant to evolutionary epidemiology. However, critical issues limit its clarity and robustness:

1. Insufficient justification of model assumptions (e.g., additivity vs. non-additivity of NFDS).

2. Ambiguity in parameter choices and simulation setups (e.g., competition parameters k values, recombination rates, initial genotype frequencies).

3. Limited exploration of alternative explanations for LD patterns (e.g., demographic effects, hitchhiking, clonality).

4. Data analysis lacks transparency in statistical testing, null model assumptions, and robustness across populations.

5. Overinterpretation: the empirical results are suggestive but not conclusive about emergent epistasis.

Major Critiques:

1. Abstract: Claims strong support for emergent epistasis from genomic data, yet the empirical evidence is largely correlative. This overstates certainty.

2. Introduction: While antigen-specific immunity is mentioned, the transition to metabolic and competition–colonisation models feels abrupt. The rationale for focusing on these two mechanisms should be justified more strongly with literature. Moreover, the introduction does not sufficiently discuss alternative balancing mechanisms (e.g., fluctuating selection, spatial heterogeneity, migration).

3. Methods, Epidemiological Models: Model assumptions are critical but under-justified. Why assume symmetric bi-allelic loci? Why ignore asymmetry in gene gain vs. loss (noted later in Supplementary but not addressed in main text)? Equations (5) and (8): Terms governing epistasis depend sensitively on parameter geometry (k0, k1, k2, kΔ). How were these parameter ranges chosen? Are they biologically realistic? Moreover, recombination treatment (σ terms) is simplistic. Empirical pneumococcal recombination is asymmetric, biased toward deletions. The model omits this key factor despite its known importance.

Phenomenological baseline (Eq. 1): assumptions, units, regime of validity: Eq. (1) adds one-locus NFDS terms to logistic growth to show that with recombination there is no sustained LD because w_AB = 0 at the allele optimum. However, the conditions under which Eq. (1) approximates the mechanistic models (weak selection, quasi-linkage equilibrium, additive fitness, timescale separation) are not stated. Please add an explicit 'Assumptions & Units' paragraph and a parameter table (confirm r, ρ have units 1/time; p is unitless).

4. Results, Simulation Evidence: Figure 2b/2c & Figure 3b/3c: The authors claim PFDS vs. NFDS regimes emerge depending on parameter geometry, but it is unclear how initial genotype frequencies (D′ = 0.4) were selected. Different starting conditions may bias toward observing bistability. In addition, figure S7/S8: Exploration of parameter space is extensive, but the biological interpretation of "concavity" and "step" parameters is not intuitive. The authors should connect these meta-parameters explicitly to measurable biological quantities and this earlier on in the texts for benefit of readership…

5. Results, Genomic Data Analysis: LD analysis is interesting but the statistical testing is insufficiently detailed. The permutation null assumes average LD = 0, which is unrealistic under recombination–selection balance. More sophisticated nulls (e.g., coalescent simulations incorporating recombination rates, clonality, demography) should be considered. Moreover, the evidence for LD sign reversals (PFDS) is rare (2.7~4.5% of pairs?). Yet this is central to the argument. The rarity of reversals suggests that constant epistasis and demographic structure may still explain most observations.

6. Discussion: The discussion overstates the empirical support for frequency-dependent epistasis. At minimum, claims should be tempered: the data are "consistent with" rather than "demonstrating" the mechanism. Also, while limitations are acknowledged, they are not sufficiently addressed. For example:

a. No discussion of recombination rate heterogeneity across populations.

b. Lack of validation against longitudinal data beyond the Massachusetts vaccine rollout.

c. No comparison to simpler explanations such as persistent clonal structure.

7. Overall Replicability: The ODE systems are detailed, but the parameterization is not fully transparent. Supplementary material provides equations but not always parameter ranges or biological justification. Furthermore, the Zenodo link to the cited code appears broken, making it impossible to verify or reproduce the simulation workflow directly. As a result, the manuscript text alone does not provide sufficient information to ensure full reproducibility without access to external files.

Minor Critiques:

1. Abstract, "emerging epistasis … unlike classical epistasis" Needs clearer definition; the distinction between frequency-dependent and classical epistasis is not well explained.

2. Results, Figure 2 captions: Ambiguity in describing "competition efficiencies"—should be defined consistently.

3. Methods: Clarify whether allele frequencies were initialized deterministically or stochastically in simulations.

4. The main text should summarize key points instead of relegating critical assumptions (like recombination asymmetry) to Supplementary.

Issue of Grammar, Spelling, Proofreading, etc.

1. Generally well written, but several sentences are long and convoluted (e.g., Discussion pp. 13–14). Shorter, more direct sentences would aid readability.

2. Inconsistent hyphenation: “co-colonisation” vs. “co colonisation”.

3. Abbreviation NFDS/PFDS should be redefined in the Discussion for clarity to non-specialist readers.

4. Typo in Eq. 9.

Reviewer #3: This manuscript addresses how multilocus negative frequency-dependent selection (NFDS) can generate emergent epistasis that, in turn, structures bacterial populations into strains. By analyzing two mechanistic models (metabolic niches and competition–colonization trade-offs), the authors show that both frequency-independent and frequency-dependent forms of epistasis can arise, including positive frequency-dependent selection (PFDS). They then test these predictions with large-scale genomic data using linkage disequilibrium (LD) analyses of accessory genes. The study is ambitious, broad in scope, and effectively bridges theoretical and empirical approaches, making it well suited for PLOS Computational Biology.

Major Comments

In the analysis of large-scale genomic data of S. pneumoniae, the identification of PFDS relies on detecting LD sign reversals across populations, under the assumption that stationary epistasis would not lead to such reversals. However, other processes (e.g., lineage-specific recombination rates, mobile genetic elements, historical bottlenecks/founder events) could also generate sign reversals. The authors should explain in greater detail how these alternative mechanisms can be excluded.

The permutation scheme shuffles gene presence/absence across strains while preserving frequencies, but this procedure destroys phylogenetic structure and signals of clonal expansion. Alternative null models—such as block permutations within lineages or phylogeny-based simulations (e.g., PAST-ML–consistent evolutionary histories)—should be considered.

While the manuscript addresses an interesting and timely question, I find the current evidence insufficient to support the central claim that PFDS and NFDS emergently structure bacterial strain diversity. The modeling framework is overly simplified to capture the complexity of pneumococcal accessory genome dynamics, and the empirical LD analyses are heavily confounded by MGEs and population structure. The inference of PFDS from rare LD reversals is particularly weak and may reflect artifacts of the null model rather than true biological signals. A more refined discussion of the gap between theoretical models and empirical data is needed.

Minor Comments

i) There appear to be discrepancies between figure references and the text. For example, in line 175, the reference to “(Figure 2a, dashed line)” seems to correspond to the red line rather than a dashed one.

ii) In line 312, the reference to “(Figure 4d)” does not correspond to an existing figure.

**Have the authors made all data and (if applicable) computational code underlying the findings in their manuscript fully available?**

Reviewer #1: Yes

Reviewer #2: Yes

Reviewer #3: None

PLOS authors have the option to publish the peer review history of their article (what does this mean?). If published, this will include your full peer review and any attached files.

Reviewer #1: No

Reviewer #2: No

Reviewer #3: No

**Figure resubmission:**
---

## [Decision Letter · Decision Letter 1]

2 Dec 2025

PCOMPBIOL-D-25-01407R1

Epistasis mediates the role of negative frequency-dependent selection in bacterial strain structure

PLOS Computational Biology

Dear Dr. Guillemet,

Thank you for submitting your manuscript to PLOS Computational Biology. After careful consideration, we feel that it has merit but does not fully meet PLOS Computational Biology's publication criteria as it currently stands. Therefore, we invite you to submit a revised version of the manuscript that addresses the points raised during the review process.

We look forward to receiving your revised manuscript.

Kind regards,

Tommy Tsan-Yuk Lam, Ph.D.

Academic Editor

PLOS Computational Biology

Jennifer Flegg

Section Editor

PLOS Computational Biology

**Reviewers' comments:**

Reviewer's Responses to Questions

**Comments to the Authors:**

Reviewer #1: [1] The authors ask, “we wonder if you would agree that the genotype frequency-dependent aspect of the epistasis is an emerging property?” I think this is reasonable, as it emerges from an interaction in the simulations, not an explicit definition within the model.

[2] I thank the authors for introducing the correction for physical linkage shown in Figures S16 and S17. These figures should replace Fig. 4a and 4b, because the comparison demonstrates that the vast majority of gene pairs showing consistent linkage between populations are close together, which is not relevant to the effect of epistasis on population structure. Similarly, S15 should be dropped, and S18 recalculated using the distance thresholds. Only these figures that incorporate the physical distance thresholds enable the reasoning below, which is detailed such that the authors can correct me if I have misunderstood:

(a) The authors state in the manuscript “the strong linkage is conserved across populations (defined as ∣D′∣ > 0.6 and same sign LD) for 30.6% of gene pairs” (this should be updated in a revision). In Fig. 4b, the shuffled population has a mode around 0.6, which is consistent with there being zero LD, as expected in these permuted data (the comparison is between D’> 0.6 in one population, and D ~ 0 in the shuffled data). The mass of smaller delta D’ values in the observed data was used as evidence of widespread epistasis in the original submission.

(b) However, 23.9% of gene pairs have to be removed through close physical linkage – similar to the proportion that exhibit strong linkage conservation across populations. Accordingly, Fig. S16 shows that the mode for the observed data delta D’ is also now 0.6 – that is, strong linkage pairs in the focal population are generally not linked in the comparator population. Therefore linkage that is observed in one population is not apparent in another. This is not consistent with widespread stable epistasis across populations. This should be made explicit.

(c) Nevertheless, the authors claim that Table S3 (note this table has this symbol in the legend: “¿100 kb”) shows that there are significantly more LD conservation events than expected between populations. This is based on the tail of small delta D values in Fig. S16, that lie outside of the narrower shuffled population distribution used as the null. However, the reason for this difference is not a difference in the position of the mode, it is a difference in the variance of the null and observed distributions. The larger variance of the observed data is due to the effects of population structure that are not accounted for in the permutations: permutation treats each isolate as an individual sample, rather than one that shares gene content through ancestry with other isolates. Hence the effective sample size of the observed population is smaller than the sample size of the shuffled population (i.e. why effective population sizes are smaller than census population sizes). Therefore the variance of the null distribution shrinks relative to the observed population.

(d) The problem this creates is best illustrated with the Maela-Southampton comparison (Table S3):

(i) There are three time more strong LD pairs in the Maela population than the Southampton one

(ii) There are the same number of observed conservation events in the comparison between the two (which is necessarily the case)

(iii) The number of expected conservation events is 2,500-fold higher for the Southampton-Maela comparison than the Maela-Southampton comparison, despite there being more strong LD pairs in Maela.

(iv) A comparison with Table S1 shows that these differences are not down to chance fluctuations in the permutation distribution – there is a 1,600-fold difference in the expected numbers for the Southampton-Maela comparison compared to the Maela-Southampton in that analysis as well

(v) After trying to work this out for some time, I found the size of the sample from Maela is much larger than the size of the sample from Southampton (https://microreact.org/project/multilocusNFDS) – this information should be reported in the manuscript – narrowing the variance of the null distribution, resulting in a reduced number of expected conservation events

(vi) Yet Fig. S16 shows the observed delta D’ distribution is approximately consistent across all population comparisons, regardless of sample size – indicating that the difference between the observed and expected values should not be strongly affected by sample size, if the comparison were appropriately calibrated

(e) This also affects the number of reversals (Table S4), which corresponds to the tail of delta D’ values >1.2. The expected number is again lowest in the comparisons using the Maela population as the focus, suggesting the effects are driven by sample size, rather than epistasis.

The authors claim that their use of genes that have been gained or lost 30 times or more will eliminate the effects of clonality. However, https://microreact.org/project/multilocusNFDS shows there are >70 Sequence Clusters, so a gene could easily be conserved within half of these (and absent from the other half), while also being gained/lost more than 30 times. For a more convincing statistical analysis:

(i) Sequence clusters observed in both compared populations (e.g. Sequence Cluster 18, found in all three) should be dropped from the comparisons, as their presence makes populations similar through common ancestry

(ii) The observed distribution of gain/loss numbers should be shown to justify the selected threshold (or it should be adjusted, as appropriate)

(iii) Null simulations should be used that replicate this gain/loss distribution – this can be done using the tree and population genetics simulation software (e.g. https://academic.oup.com/sysbio/article/65/2/334/2427219)

(iv) D’ values from these null simulations (with genes having similar gain/loss distributions in the observed and simulated populations) can then be compared with the observed ones as in the current manuscript

All these problems also affect the pre/post vaccine analysis of the Massachusetts population. The current version is making claims that simply cannot be robustly justified (e.g. much of the Discussion). While I appreciate the authors have added some caveats, they are not sufficient to highlight the problems to many readers with a biological background.

Additionally, what is the explanation for the atypically strong associations seen with Translation genes in Fig 4a/S17? Is this down to a small number of genes, as translation machinery should rarely be in the accessory genome?

[3] Thank you for these additions – Fig. S2 needs a legend.

Reviewer #2: The comments and concerns raised have been addressed.

Reviewer #3: The authors have satisfactorily addressed all previous comments and concerns. The revisions have significantly improved the clarity and quality of the manuscript. I am satisfied with the current version and recommend acceptance in its present form.

**Have the authors made all data and (if applicable) computational code underlying the findings in their manuscript fully available?**

Reviewer #1: Yes

Reviewer #2: None

Reviewer #3: None

PLOS authors have the option to publish the peer review history of their article (what does this mean?). If published, this will include your full peer review and any attached files.

Reviewer #1: No

Reviewer #2: No

Reviewer #3: No

**Figure resubmission:**
---

## [Decision Letter · Decision Letter 2]

5 Feb 2026

PCOMPBIOL-D-25-01407R2

Epistasis mediates the role of negative frequency-dependent selection in bacterial strain structure

PLOS Computational Biology

Dear Dr. Guillemet,

Thank you for submitting your manuscript to PLOS Computational Biology. After careful consideration, we feel that it has merit but does not fully meet PLOS Computational Biology's publication criteria as it currently stands. Therefore, we invite you to submit a revised version of the manuscript that addresses the points raised during the review process.

We look forward to receiving your revised manuscript.

Kind regards,

Tommy Tsan-Yuk Lam, Ph.D.

Academic Editor

PLOS Computational Biology

Jennifer Flegg

Section Editor

PLOS Computational Biology

**Journal Requirements:**

**Reviewers' comments:**

Reviewer's Responses to Questions

**Comments to the Authors:**

Reviewer #1: I thank the authors for their patience, and the changes they have made to the analysis, which address my concerns.

My only remaining comments regard the presentation of the Results. The authors state that “The evidence for LD conservation is less clear (Table 2): we find patterns of LD being both more and less conserved than expected by chance.” As it is unlikely that simulations can exactly replicate the observed level of LD, this result essentially implies that the observed level is indistinguishable from that expected by chance, and therefore there is no significant evidence of LD.

The authors state in the Results, “This filtering is likely to remove some pairs of genes in strong positive epistasis (e.g. genes within the same operon),” then in the Discussion they state, “the removal of physically close gene pairs underestimates the extent of positive epistasis.” Similarly, elsewhere in the revised Results it is stated that “the distance-based filtering we use to limit the impact of physical linkage is very likely to filter out gene pairs in strong positive LD, which would contribute to the lack of conservation we observe.”

However, such short-range interactions do not affect the structure of the population, because each of the genes will be co-transferred by individual horizontal gene transfer events. The authors themselves acknowledge this in the Introduction, which states, “Epistasis refers to the interaction between genes at different loci.”

The authors should change the Results and Discussion to reflect the lack of evidence for classical epistasis shaping the population structure, as there is no evidence for long-range epistasis being conserved across populations.

Additionally, in the Abstract, it is stated, “make observations consistent with epistatic effects on gene associations”. This would be interpreted by a casual reader as evidence of classical epistasis. It would be helpful to clarify the findings by editing the statement to “make observations consistent with frequency-dependent epistatic effects on gene associations.”

One additional comment: the text refers to “clonal genealogies.” However, it is not clear that recombination could have been removed when constructing such a phylogeny, so I suspect it is not “clonal,” but an approximation from an alignment of recombining genotypes.

Reviewer #2: No further comments.

Reviewer #3: The authors have satisfactorily addressed all previous comments and concerns.

**Have the authors made all data and (if applicable) computational code underlying the findings in their manuscript fully available?**

Reviewer #1: Yes

Reviewer #2: Yes

Reviewer #3: None

PLOS authors have the option to publish the peer review history of their article (what does this mean?). If published, this will include your full peer review and any attached files.

Reviewer #1: No

Reviewer #2: **Yes:** Rajib Saha

Reviewer #3: No

**Figure resubmission:**
---

## [Decision Letter · Decision Letter 3]

3 Mar 2026

Dear M. Guillemet,

We are pleased to inform you that your manuscript 'Epistasis mediates the role of negative frequency-dependent selection in bacterial strain structure' has been provisionally accepted for publication in PLOS Computational Biology.

Best regards,

Tommy Tsan-Yuk Lam, Ph.D.

Academic Editor

PLOS Computational Biology

Jennifer Flegg

Section Editor

PLOS Computational Biology

Reviewer's Responses to Questions

**Comments to the Authors:**

Reviewer #1: I thank the authors for their patience with my comments.

Reviewer #2: It seems the comments raised were addressed.

Reviewer #3: There are no additional comments.

**Have the authors made all data and (if applicable) computational code underlying the findings in their manuscript fully available?**

Reviewer #1: Yes

Reviewer #2: None

Reviewer #3: None

PLOS authors have the option to publish the peer review history of their article (what does this mean?). If published, this will include your full peer review and any attached files.

Reviewer #1: No

Reviewer #2: **Yes:** Rajib Saha

Reviewer #3: No